# CXCR4/CXCL12 Activities in the Tumor Microenvironment and Implications for Tumor Immunotherapy

**DOI:** 10.3390/cancers14092314

**Published:** 2022-05-06

**Authors:** Rosanna Mezzapelle, Manuela Leo, Francesca Caprioglio, Liam S. Colley, Andrea Lamarca, Lina Sabatino, Vittorio Colantuoni, Massimo P. Crippa, Marco E. Bianchi

**Affiliations:** 1School of Medicine, Vita-Salute San Raffaele University, 20132 Milan, Italy; mezzapelle.rosanna@hsr.it (R.M.); caprioglio.francesca@hsr.it (F.C.); lamarca.andrea@hsr.it (A.L.); 2Division of Genetics and Cell Biology, IRCCS San Raffaele Hospital, 20232 Milan, Italy; crippa.massimo@hsr.it; 3Department of Sciences and Technologies, University of Sannio, 82100 Benevento, Italy; manuela.leo@unisannio.it (M.L.); sabat@unisannio.it (L.S.); colantuoni@unisannio.it (V.C.); 4HMGBiotech S.r.l., 20133 Milan, Italy; liam.colley@hmgbiotech.eu; 5School of Medicine and Surgery, Università Milano-Bicocca, 20126 Milan, Italy

**Keywords:** immunotherapy, ImmunoGenic Surrender, CXCL12, CXCR4, ACKR3, CD47, BCR, TCR

## Abstract

**Simple Summary:**

Chemokines are small soluble proteins that control and regulate cell trafficking within and between tissues by binding to their receptors. Among them, CXCL12 and its receptor CXCR4 appeared with ancestral vertebrates, are expressed almost ubiquitously, and play essential roles in embryogenesis and organogenesis. In addition, CXCL12 and CXCR4 are involved in antigen recognition by T and B cells and in shaping the tumor microenvironment (TME), mainly towards dampening immune responses. New data indicate that CXCR4 interacts with the surface protein CD47 in a novel form of immunosurveillance, called ImmunoGenic Surrender (IGS). Following the co-internalization of CXCR4 and CD47 in tumor cells, macrophages phagocytose them and cross-present their antigens to the adaptive immune system, leading to tumor rejection in a fraction of mice. All of these specific activities of CXCL12 and CXCR4 in antigen presentation might be complementary to current immunotherapies.

**Abstract:**

CXCR4 is a G-Protein coupled receptor that is expressed nearly ubiquitously and is known to control cell migration via its interaction with CXCL12, the most ancient chemokine. The functions of CXCR4/CXCL12 extend beyond cell migration and involve the recognition and disposal of unhealthy or tumor cells. The CXCR4/CXCL12 axis plays a relevant role in shaping the tumor microenvironment (TME), mainly towards dampening immune responses. Notably, CXCR4/CXCL12 cross-signal via the T and B cell receptors (TCR and BCR) and co-internalize with CD47, promoting tumor cell phagocytosis by macrophages in an anti-tumor immune process called ImmunoGenic Surrender (IGS). These specific activities in shaping the immune response might be exploited to improve current immunotherapies.

## 1. Introduction

Chemokines are small soluble proteins that control and regulate cell trafficking within and between tissues through binding to their receptors [1]. This function is now well appreciated and has been reviewed in depth [2,3,4].

Chemokines can be grouped into four families (CC, CXC, CX3C, and XC) based on the arrangement of the two cysteine residues near the amino terminus (C is cysteine and X any other amino acid; X3 means three consecutive amino acids).

Chemokine receptors are usually G protein-coupled receptors (GPCRs), containing seven transmembrane alpha-helices and six loops (three extracellular and three intracellular); the N-terminus is located outside the cell and the C-terminus in the cytoplasm. They are divided into two subfamilies: conventional chemokine receptors (cCKR) and atypical chemokine receptors (ACKR) [5]; both undergo conformational changes upon ligand binding that allow dissociation of the G proteins. Conventional chemokine receptors contain a conserved DRYLAIV motif, while ACKRs contain a modified version [6].

Both chemokines and their receptors arose by repeated gene duplication in the vertebrate lineage [7], and members of each family are structurally related. As a consequence, communication between chemokines and chemokine receptors is highly redundant: many chemokines can bind more than one receptor and, in turn, chemokine receptors can bind multiple chemokines. The complexity of chemokine-receptor interactions is further increased by the fact that chemokines form homomeric and heteromeric complexes among themselves and other proteins, and each complex can be thought of as an additional ligand [8]; moreover, the functional properties of chemokine receptors are modulated by the interaction with other GPCRs and membrane lipids [9]. Chemokine receptors regulate many functions, such as immunity and embryonic development, by directing cellular migration, adhesion, growth, and survival.

We will focus here on CXCR4 and its ligand CXCL12, the most ancient receptor-ligand pair, orthologues of which are present in the sea lamprey [7]. In particular, after a short recapitulation of background information, we will discuss the activities of CXCR4 and CXCL12 in the tumor microenvironment and the ability of CXCR4 to interact laterally with other receptors and to modulate processes that can be exploited in the context of immunotherapy.

## 2. CXCL12 and Its Receptors CXCR4 and CXCR7/ACKR3

CXCL12 (also known as Stromal Cell-Derived Factor-1, SDF-1) is widely expressed in different tissues and is the only chemokine essential for embryo development and organogenesis [10]. CXCL12 binds to the CXC chemokine receptor 4 (CXCR4, also known as CD184) and to CXCR7/ACKR3 [11]. CXCL12 and CXCL14 are the only chemokine ligands for CXCR4; CXCL14 modulates CXCL12 bioactivity without having chemotactic activity by itself [12].

The *CXCL12* gene in humans, located on chromosome 10q11, is transcribed and subject to alternative splicing of the third and fourth exons, giving rise to six differentially expressed proteins (CXCL12 α to θ) [13]. The orthologous *Cxcl12* gene in mice produces three differentially expressed proteins (CXCL12 α to ϕ). The various isoforms have differential affinity for glycosaminoglycans on the cell surface and in the extracellular matrix [14].

CXCL12 belongs to the family of CXC chemokines, which are sub-classified based on the presence or absence of the ELR (Glu-Leu-Arg) motif immediately before the CXC motif. ELR-positive CXC chemokines have a proangiogenic function, while most ELR-negative CXC chemokines are angiostatic [15]. However, both CXCL12 and CXCL2 are ELR-negative but proangiogenic; indeed, CXCL12 is one of the most potent angiogenesis-promoting chemokines [16,17].

CXC chemokines are also classified as inflammatory, homeostatic, or dual function. CXCL12 is a homeostatic chemokine that can exist in monomeric and dimeric forms [18]; it has key roles in embryogenesis, angiogenesis, lymphopoiesis and regulates hematopoietic stem cell trafficking in physiological and pathological conditions [12,19,20].

CXCR4 is a 352 amino acid GPCR that can exist as a monomer, dimer, oligomer, or nanoclusters in the plasma membrane [21].

Binding of CXCL12 to CXCR4 is followed by phosphorylation of its intracellular C-terminal domain at multiple sites, resulting in the dissociation of the Gβγ and Gα subunits and activation of the MAPK, PI3K and phospholipase C pathways (Figure 1). This signaling (reviewed in Ref. [22]) leads to intracellular calcium mobilization and cell proliferation, differentiation, migration, and survival; notably, cell proliferation and cell migration require the activation of the same pathways, although proliferation and migration are mutually exclusive processes. Binding of CXCL12 to CXCR4 also induces β-arrestin recruitment, which in turn can signal via the MAPK p38 [22,23] and induce clathrin-dependent internalization of the whole receptor-ligand complex [24].

The structure of CXCR4 bound to CXCL12 has been investigated extensively, and a detailed model integrating crystallography and mutagenesis is available [25].

## 3. Heteromeric Interactions of CXCR4 with ACKR3 and Other Chemokine Receptors

Atypical chemokine receptor 3 (ACKR3) is expressed in mesenchymal stromal cells, in cells of the central nervous system, including astrocytes, glial and neuronal cells, in vascular smooth muscle cells and the venous endothelium, and in immune cells: B cells and plasmablasts, but not mature plasma cells, natural killer (NK) cells, basophils, dendritic cells (DCs) and CD4 but not CD8 T cells [26].

ACKR3 was initially identified as a decoy chemokine receptor for CXCL11 and CXCL12. Two substitutions (A/S and V/T) in the DRYLAIV motif preclude G protein activation and intracellular Ca^2+^ release [27], even though β-arrestin2 can still be recruited. For this reason, it was initially thought that ACKR3 was a nonsignaling receptor. Indeed, in zebrafish, ACKR3 acts as a scavenger receptor to internalize and eliminate CXCL12, thereby helping to create a CXCL12 gradient that guides the migration of germ cells [28]; ACKR3 helps reduce CXCL12 levels in mammalian cells as well [29,30]. However, ACKR3 does signal by itself [26,31] and does modulate the expression and activity of CXCR4 and CXCL12 through a variety of mechanisms. First, CXCL12 shows higher affinity for ACKR3 than CXCR4, but both association and dissociation rates of CXCL12 with ACKR3 are slower than those with CXCR4, thus making CXCR4 kinetically favored for CXCL12 binding [32]. Moreover, ACKR3 and CXCR4 can form heterodimers (Figure 1) where signaling though β-arrestins prevails over G-protein signaling [33,34,35].

CXCR4 forms heteromers not just with ACKR3, though. In leukocytes, it forms heteromers with chemokine receptors CCR2, CCR7, CCR5, and CXCR3. CCR2 is the receptor for CCL2/MCP-1 (Monocyte Chemotactic Protein 1), and their interaction is the main driver of the recruitment of monocytes to tissues, including cancer tissues. In CXCR4-CCR2 heteromers, ligands cooperate negatively and antagonists cross-inhibit ligand binding to the other interacting receptor [36,37]. CCR5 is the receptor for CCL5/RANTES, a chemokine produced by T cells, macrophages and activated platelets and which chemoattracts monocytes, memory T helper cells and eosinophils. CCR7 not only forms heteromers with CXCR4, but also requires CXCR4 for proper expression and activity on the surface of T cells [38]. CXCR4 signaling facilitates CCR7 ligand binding and increases the level of CCR7 homo- and CXCR4/CCR7 heteromers without affecting CCR7 expression levels.

CXCR4 can also heteromerize with other GPCRs, such as adrenergic and δ and κ opioid receptors, and may contribute to blood pressure and pain modulation [39,40,41,42].

## 4. CXCL12/CXCR4/ACKR3 and Tumors

The role of CXCL12/CXCR4 in promoting the migration of cells to developing tissues and organs during organ development and tissue repair has been widely reviewed [22,43]. However, they also play a key role in the establishment and maintenance of tumor tissues, and in particular in their microenvironments. Microenvironment (ME) is the term used to indicate the milieu formed by cells and stromal components as well as locally produced or conveyed molecules. The ME ultimately dictates the fate and the activity of the resident cells. Tumor microenvironment (TME) indicates the specific milieu that in cancer tissues surrounds parenchymal tumor cells and is composed by microvessels, immune cells, stromal fibroblasts, endothelial cells, growth factors, inflammatory cytokines, proteolytic enzymes and the extracellular matrix [44].

CXCL12 and CXCR4 within the TME generally promote tumor angiogenesis and the survival and proliferation of tumor cells; they also recruit immune cells and direct them to specific immunosuppressive responses [45].

Macrophages constitute a prominent fraction of infiltrated leukocytes within the TME [46,47]. Tissue macrophages display a remarkable functional and phenotypic diversity that allows them to proceed from an uncommitted state to a pro-inflammatory state (often called M1) or to a tissue-healing and -remodeling state (M2) [48]. The adoption of a specific phenotypic state is called polarization and is promoted by environmental cues from the TME [49,50]. Consistently, the M1/M2 ratio decreases with cancer stage; the prevalence of M2 macrophages has been correlated with a worse clinical outcome in ovarian cancer and a series of other malignancies [51,52]. CXCL12 promotes the recruitment of Tumor Associated Macrophages (TAMs), which usually adopt the M2 phenotype and favor tumor progression [51,53].

Tumor cells in advanced cervical cancer [54], malignant pleural mesothelioma [55], ovarian cancer [56] and renal cell carcinoma [57] secrete CXCL12, which recruits both TAMs and T-regulatory cells (Tregs) via binding to CXCR4 expressed at high levels on their surface. Tregs dampen effector immune cells’ functions in various ways (via cytokines, cell lysis, inhibitory receptors, metabolic reprogramming), thus favoring cancer progression. Consistently, treatment of mice bearing ovarian tumors with the CXCR4 antagonist AMD3100 conferred a survival advantage by reducing intratumoral Tregs and converting them into T-helper-like cells [56].

In contrast to Tregs, the proportion of CD8 T cells in the TME correlates with a good prognosis in breast, colorectal, glioblastoma and cervical cancers. In fact, upon exposure to specific signals in the TME, naïve CD8 T cells differentiate into effector T cells and further into cytotoxic and memory CD8 T cells [58], but CXCL12 hampers their differentiation into effector cells. Both naïve and memory CD8 T cells in the mouse bone marrow express CXCR4 at high levels, which regulates bone marrow-homing [59] as well as migration towards vascular CXCL12-positive cells.

Tumor cells also recruit myeloid derived suppressor cells (MDSCs) via CXCL12-CXCR4 signaling; MDSCs suppress anti-tumor immune responses and enhance tumor growth, for example in (but not limited to) osteosarcoma [60].

Dendritic cells (DCs), the antigen-presenting cells (APC) with a crucial role in the adaptive immune response, are also engaged by TME-produced CXCL12. DCs express CXCR4 at high levels starting from the immature stage in the bone marrow so they can reach peripheral tissues or the TME. Here, CXCL12 reduces their efficiency as APCs and decreases their immune response, facilitating tumor growth.

CXCL12 interacts also with the CXCR4 present on the surface of mature B cells and mediates the recruitment of B regulatory cells to the tumor site inhibiting T cell activity [8]. Indeed, only a few B cells are detected in tumor infiltrates, where they may exert both pro-tumor and anti-tumor effects according to the TME, their phenotypes and antibodies production [61]. B cells express CXCR4 at all stages during their development in bone marrow, with a crucial role in the homing of B cell precursors [62]. CXCR4 is thus necessary for the B cell development but not for mature B cells [63,64]. In a murine breast cancer model, primary tumors induce B cell accumulation in draining lymph nodes and foster metastasis through activation of the CXCL12/CXCR4 axis [65]. B cells may negatively regulate tumor immunity and promote tumor progression via IL-10 and TGF-β expression [66]. Mature B cells in the splenic marginal zone also express ACKR3, which is involved in the regulation of their development and differentiation [67,68].

In addition to tumor cells, cancer-associated fibroblasts (CAFs) can synthesize and secrete CXCL12. These cells engage and activate CXCR4 expressed on the surface of breast cancer cells [69] to promote tumor growth and invasiveness. Furthermore, CAF-secreted CXCL12 recruits more CXCR4 expressing cells, like endothelial cells, epithelial cells and immune cells to activate intracellular pathways leading to cell proliferation and tumor progression [70]. Diverse subsets of CAFs characterize high grade serous ovarian cancers (HGSOC); in particular, the mesenchymal HGSOC, a molecular subgroup with poor patient prognosis, have a high content of the CAF-S1 myoblast subset that expresses high levels of CXCL12. This is essential for T cell attraction towards the HGSOC and for immunosuppression. CAF-S1 enhance the survival as well as the activation of regulatory T lymphocytes, independently of CXCL12, which could account for the poor survival of mesenchymal HGSOC patients [71].

Endothelial cells express both CXCR4 and ACKR3 receptors and secrete CXCL12, which facilitates the trafficking of cancer and immune cells between the bloodstream and the adjacent tissues [72]. Furthermore, CXCR4 promotes angiogenesis by recruiting endothelial progenitor cells or bone marrow-derived accessory cells and, through VEGF, stimulates sprouting angiogenesis [73]. ACKR3 is highly expressed, relative to the normal vasculature, by most tumor-associated blood vessels of human breast and lung cancers, as well as of melanoma, modulating tumor associated angiogenesis [74].

Overall, while the recruitment of immune cells to the TME via the CXCL12/CXCR4 axis is reasonably well understood, the mechanisms whereby they are polarized or differentiated to a state with low antitumor responsiveness is less well understood.

## 5. CXCL12 Promotes Immune Surveillance by Macrophages

Although the CXCL12/CXCR4 axis is directly or indirectly tumor-promoting and immunosuppressive, we have recently uncovered a counterbalancing activity: the promotion of cell phagocytosis by macrophages via a lateral interaction with CD47 on the cell to be ingested, that ultimately promotes the emergence of anti-tumor effector T cell clones [75].

CD47 (also called Integrin Associated Protein, IAP) is a ubiquitously expressed surface glycoprotein of the immunoglobulin superfamily, consisting of a heavily glycosylated extracellular domain (containing a disulfide-bonded IgV-like domain), five transmembrane (TM) domains and a short cytoplasmic tail ranging from three to 36 amino acids. Four isoforms have been identified that derive from alternative splicing and are expressed in different tissues [76]. An additional disulfide bridge also forms between the extracellular domain and one of the TM domains.

CD47/IAP was discovered originally as a plasma membrane molecule that copurifies with the integrin αvβ3 [77]; in fact, it interacts via its Ig-like domain with multiple integrins, including αIIbβ3 and α2β1. The interaction with integrins is in cis, i.e., between molecules in the same membrane. CD47 also interacts with thrombospondins, a family of five glycoproteins regulating cell migration, proliferation, differentiation and inflammation [78]. Interactions with both integrins and thrombospondins activate G_i_ protein signal transduction.

Although CD47 does not appear to interact with cytoplasmic proteins, lateral interactions with other receptors can allow its downstream signaling. Besides specific integrins, CD47 directly associates with vascular endothelial growth factor receptor-2 (VEGFR2) in endothelial and T cells, and TSP-1 binding abrogates VEGFR2 signaling [79]. CD47 also interacts with the activated Fas receptor, promoting Fas-mediated apoptosis [80]. Lateral interaction between CD47 and CD14 occurs in human resting but not LPS-stimulated monocytes [81].

In 1999, CD47 was found to interact with SIRP proteins, a family of cell surface glycoproteins belonging to the Ig superfamily [82]. Importantly, the crystal structure of the CD47-SIRPα complex has been resolved [83], and their interactions are well understood. SIRPα is expressed by myeloid cells, in particular macrophages, neurons, endothelial cells and fibroblasts. The binding of CD47 to SIRPα (and γ) is in trans (i.e., the two molecules are on the surface of different cells) and mediates cell-cell adhesion, is highly species-specific [82], and activates signaling that inhibits macrophage phagocytosis. Hence, CD47 works as a signal of “self” and its expression on healthy cells prevents their removal; cells with low CD47 are phagocytosed, which is crucial in the maintenance of tissue integrity and homeostasis.

CD47 is expressed at a high level on the cell surface by a variety of malignant cells; notably, its blockade with monoclonal antibodies allows the efficient phagocytosis of cancer cells and leads to tumor rejection and the development of antitumor immunity [84]. For this reason, many antibodies and antagonists targeting the CD47-SIRPα interaction (or “macrophage immune checkpoint”) have been designed and tested. While monotherapy with the CD47 blockade shows efficacy in several syngeneic mouse models of cancer, it proved ineffective in clinical trials so far, except for rare cutaneous and peripheral lymphomas [85]. However, combination therapy in humans of anti-CD47 and anti-CD20 (rituximab) shows promising activity against non-Hodgkin lymphoma [86].

The direct phagocytosis of tumor cells by macrophages is not sufficient per se to reduce the tumor burden; tumor elimination involves the emergence of CD8 T cell clones that recognize tumor-associated antigens and kills tumor cells that express them. Therefore, it is presumed that macrophages process the tumor cells they have ingested and act as Antigen Presenting Cells to the adaptive immune system. Importantly, healthy cells are not killed by T cells after CD47 blockade, which suggests that additional microenvironmental cues allow macrophages to distinguish between tumor and healthy cells.

We have recently shown that CXCR4 interacts with CD47, in a novel lateral interaction [75] (Figure 1). Proximity Ligation Assays (PLA) indicate that CXCR4 and CD47 physically interact on the surface of tumor cells; the binding of CXCL12 to CXCR4 leads to the co-internalization of the CXCR4-CD47 complexes, and therefore to the decrease of surface CD47 (Figure 2). Interfering with CXCR4 expression via shRNA or pharmacologically with AMD3100 reduces CD47 depletion from the cell surface. CXCL12 addition to coculture of mouse mesothelioma tumor cells and bone marrow derived macrophages leads to the phagocytosis of a substantial fraction of tumor cells.

CXCL12 addition to tumor cells also induces the release of damage associated molecular patterns such as ATP, HMGB1 and calreticulin; in turn, HMGB1 can promote the secretion of CXCL12 [87]. Moreover, BoxA, a fragment of HMGB1 that corresponds to its first HMG domain, can also bind to CXCR4 and promote CXCR4-CD47 co-internalization and tumor cell phagocytosis [75]. Thus, the functions of HMGB1 and CXCL12 appear highly cross-dependent, which is further highlighted by the fact that HMGB1 and CXCL12 form a heterocomplex with enhanced activity on CXCR4 [88]. Notably, mesothelioma tumor cells secrete abundant HMGB1 [89,90], which may be the ultimate driver of macrophage recruitment inside the tumor. In fact, the release of DAMPs, including the eat-me signal calreticulin, may be necessary in addition to CD47 surface depletion to induce phagocytosis by macrophages, and may allow the discrimination of tumor cells from healthy cells.

## 6. The Role of CXCR4 in T Cell Receptor (TCR) and B Cell Receptor (BCR) Signaling

The functional role of CXCR4 in allowing and promoting immune cell responses extends beyond macrophages.

Remarkably, in T cells CXCR4 and the TCR/CD3 complex interact and signal through each other [91] (Figure 1); such TCR and CXCR4 cross-signaling affects T cell epitope recognition and cytokine secretion in addition to cell migration. The cross-signaling involves the TCR ITAM domains, CD3ζ, p52Shc, Lck2 and ZAP-70, and results in robust and prolonged RAS and ERK activation, which mediates AP-1-dependent gene transcription, T cell proliferation and chemotaxis [92,93], all of them essential functions of the TCR.

CXCL12/CXCR4 signaling is also necessary for TCR-initiated formation of the immune synapse: CXCR4 downregulation or blockade on T cells causes defective actin polymerization at the contact site with the antigen presenting cell (APC), altering microtubule-organizing center polarization and the structure of the immunological synapse, and reducing T cell/APC contact duration [94,95].

Conversely, TCR ligation induces the formation of the TCR-CXCR4 complex and CXCR4 transactivation, in practice allowing the TCR to signal via CXCR4. Phosphorylation of CXCR4 S339 activates the phosphatidylinositol 3,4,5-trisphosphate-dependent Rac exchanger 1 (PREX1) protein. The PREX1-Rac1 signaling pathway stabilizes IL-2, IL-4, and IL-10 mRNAs, leading to robust cytokine secretion [96,97].

CXCL12/CXCR4 is also needed for the thymic β-selection developmental checkpoint, which screens for productive rearrangement of the T cell antigen receptor-β gene (*TCRB*) and assembly of a surface pre-TCR complex. The pre-TCR regulates CXCR4-dependent migration and, reciprocally, CXCR4 influences the preTCR-dependent induction of survival signals [98].

Interestingly, CXCR4 signals through the BCR in B cells [63]. Mature B cells express two different BCRs that reside in different regions of the cell membrane; one BCR includes IgM and the other IgD membrane-bound antibodies with the same antigen specificity. CXCR4 is physically associated with the IgD-BCR, but not the IgM-BCR, and CXCL12 signals are processed through it and the associated CD19. Thus, B cell migration and the activation of PI3K/Akt/FoxO and Erk MAPK pathways in response to CXCL12 are abrogated in the absence of the IgD-BCR.

## 7. Potential Application to Immune Therapy

A role of CXCR4 in antitumor responses is unexpected, since the CXCL12-CXCR4 axis has been generally correlated with tumor cell growth, survival, invasion and metastization [98,99,100,101,102]. Indeed, the CXCL12-CXCR4 axis is involved in tissue repair and regrowth after damage [22], and may be promoting tumor cell growth in similar ways. However, as we described in the preceding sections, CXCR4 signaling has been known for some time to be intimately intertwined with TCR and BCR signaling, and thus with antigen recognition, and more recently with the singling out of cells to be phagocytosed by macrophages.

We designated “ImmunoGenic Surrender” (IGS), the mechanism whereby CXCR4 engagement drives CD47 surface depletion and eventually the appearance of tumor-specific CD8 T cell clones, as the tumor cells surrender to macrophages, which phagocytose them when still alive [75]. Most of our experiments were performed on a mouse mesothelioma model using BoxA, a fragment of HMGB1 that, like CXCL12, engages CXCR4 and induces its internalization together with CD47. However, IGS also drives the rejection of colorectal carcinoma in a fraction of mice. Notably, neither BoxA nor CXCL12 induce tumor cell apoptosis, and are not toxic to mice.

We envision IGS as a counterbalancing activity of the CXCL12/CXCR4 axis that submits the repairing/growing tissue to immunosurveillance. In fact, IGS may drive spontaneous tumor rejection. Activation of CXCR4 by administration of exogenous ligands like BoxA, which in contrast to CXCL12 does not promote cell proliferation and tissue growth, might represent a promising antitumor strategy. Notably, while targeting CD47 with monoclonal antibodies only removes the don’t-eat-me signal, reinforcement of IGS with CXCR4 ligands would provide eat-me signals on tumor cells as well.

## 8. Conclusions

The activation of macrophage phagocytosis of tumor cells and co-presentation of their antigens may synergize with the chemoattractant activity of CXCL12 towards monocytes and macrophages, with CXCR4-TCR co-signaling in the immune synapse and with BCR co-signaling in mature B cells. All of these converging mechanisms should give rise to a rich diversity of tumor-recognizing lymphocyte clones. Therefore, interest might emerge in developing agonists of CXCR4 as anti-tumor drugs, especially if the effects specific for immune cells can be separated from general effects on angiogenesis and cell proliferation. Since CXCR4 is a GPCR and relies on multiple downstream signaling pathways, it might be possible to design specific “biased” activators for its different activities (see Ref. [103] for a review on biased signaling in GPCRs). So far, promotion of autoimmune disorders via the CXCR4/CXCL12 axis has not been described, but this of course might be for lack of specific investigation. We predict that the emergence of anti-tumor T cell clones promoted by the CXCR4/CXCL12 axis should synergize with current immunotherapies, which remove the brakes of T cells once they recognize the antigen. Such speculation, however, awaits experimental testing.

## Figures and Tables

**Figure 1 cancers-14-02314-f001:**
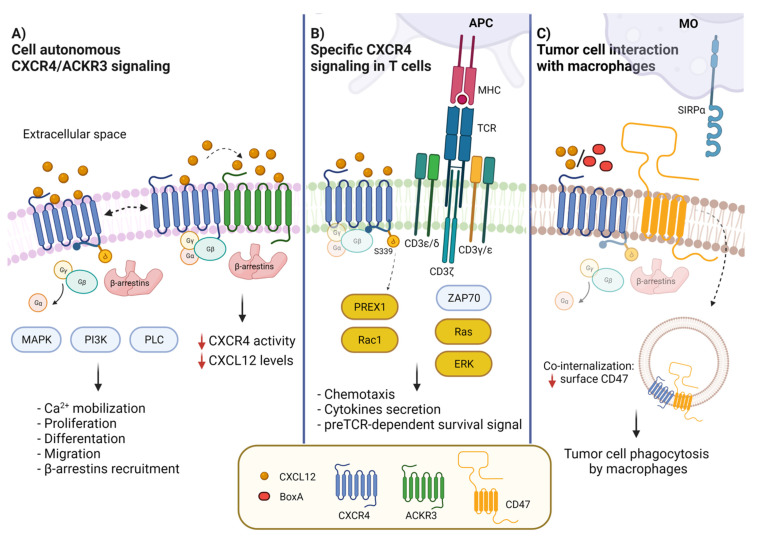
CXCR4 membrane interactors and CXCL12-activated signal transduction pathways. (**A**) CXCL12 binding to CXCR4 leads to the activation of G protein-dependent signaling, MAPK, PI3K and PLC pathways, resulting in diverse biological outcomes such as calcium mobilization, cell proliferation, differentiation, migration and adhesion. CXCR4 can form a heterodimer with ACKR3 (CXCR7), which contributes to modulate CXCL12/CXCR4 signaling. ACKR3 impairs CXCR4-mediated G-protein activation and calcium responses. (**B**) In T cells, CXCR4 interacts with CD3/TCR. The activation of CXCR4 by CXCL12 induces the formation of the immunological synapsis with antigen presenting cells (APCs) and the activation of the RAS-ERK pathway. The interaction of the TCR with CXCR4 causes the phosphorylation of CXCR4-S339, which leads to PREX-Rac1 activation and to cytokine synthesis and secretion. (**C**) CXCR4 and CD47 are in contact in cancer cells; the binding to CXCR4 of CXCL12 or BoxA, a truncated form of HMGB1, promotes co-internalization of CXCR4-CD47. Reduction of surface CD47 impairs its recognition by SIRPα, allowing tumor cell phagocytosis by macrophages.

**Figure 2 cancers-14-02314-f002:**
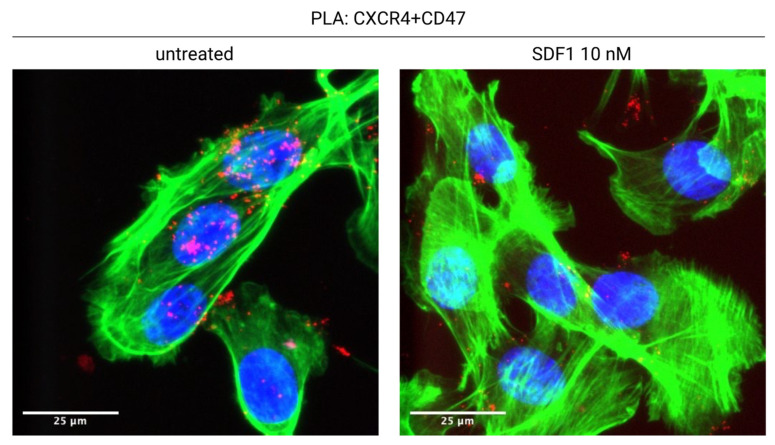
CXCL12 induces the internalization of the CXCR4-CD47 complex. CXCR4 and CD47 are in physical contact on the surface of untreated tumor cells (AB1-B/c-LUC mesothelioma cells, stained green with phalloidin; CXCR4-CD47 Proximity Ligation Assay signal, red). Overnight treatment with CXCL12 causes the almost complete disappearance of the CXCR4-CD47 signal from the cell surface. See [75] for details.

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
