# Peer review of "CXCR4/CXCL12 Activities in the Tumor Microenvironment and Implications for Tumor Immunotherapy"

_cancers, 2022, doi:10.3390/cancers14092314_

Round 1
Reviewer 1 Report
The authors have substantially improved their manuscript although they could have include more information about the canonical role of CXCL12 in chemotaxis and migration. I have no additional comments.
This manuscript is a resubmission of an earlier submission. The following is a list of the peer review reports and author responses from that submission.
Round 1
Reviewer 1 Report
In this review Bianchi and colleagues, revised the role of CXCL12/CXCR4 in physiology and cancer, particularly focused on its immunity-promoting activities. The review is interesting and easy to read. However, it seems rather short to accomplish for an accurate description of CXCL12 signaling complexity taking into account that is a review on the CXCL12/CXCR4 axis that displays several functions, in particular in the context of cancer. Considering that the title of the review is rather general, authors have not properly discussed with sufficient detail many of these aspects and focused more in just one type of function (interaction with CD47) that still requires further investigation and in which authors have been published several articles. For improving the quality of this review, please take into consideration the following comments:
1.Authors presented a clear introduction regarding the families or chemokines and receptors. They comment that ELR-positive CXC chemokines have a proangiogenic function. However, CXCL2 also shows a strong angiogenic profile without an ELR motif in their structure. Here, authors should provide details regarding this very relevant function and discussed on which structural determinants could be involved.
- Although there is an extensive description about the types of chemokines and their receptors, the signaling pathways downstream chemokines and in particular downstream CXCL12/CXCR4 are poorly described in the text and deserved more details.
- A related comment is that signaling changes in CXCL12/CXCR4 pathway occurring when ACKR3 is present require further explanations. Transcriptional regulation/inducers of ACKR3 and cell context should be discussed in order to clarify its relevance in CXCL12/CXCR4 signaling.
- The classical functions of CXCL12/CXCR4 as a chemotaxis agent are too briefly described and deserved further analysis in terms of the mechanism and in different contexts (during development, neuron migration, hematopoietic systems and stem cells). Please also include specific information about the downstream signaling leading to chemotaxis/migration. This is the most studied and canonical function of CXCL12 and deserves more description in a review dedicated to CXCL12. Authors can better introduce known inhibitors of CXCR4 in this section. Additionally, the pro-angiogenic role of this chemokine should be discussed.
- CXCL12/CXCR4 plays multiple roles in the tumor microenvironment and authors do skip this relevant part that could counterpart the pro-phagocytic role proposed by the interaction between CD47 and CXCL12/CXCR4. It would be convenient to introduce first some of the pro-tumor functions of CXCL12/CXCR4 with enough detail before focusing on the interaction with CD47.
- Conclusions are poor. Authors could summarize what they have discussed or rather proposed some specific areas that deserve further investigation as the complexity of integrate all the pro-tumor with the pro-phagocytic activity of CXCL12/CXCR4, which specific subset of macrophages is susceptible of this type of signaling, etc. Authors could also comment on the absence or presence of expected autoimmune effects. They can speculate with the possibility of used the proposed the immunogenic surrender mechanism in combination with current immunotherapies exploiting immune checkpoints (anti-PD1/PD-L1; anti-CTLA-4, etc.).
- In figure 1, Please consider to better divide/indicate the drawing in 3 panels: a, b, c. This will facilitate the interpretation through the figure legend. In figure legend, a brief description of AMD3100 is missing.
Reviewer 2 Report
Nice paper. I endorse it's publication.
Reviewer 3 Report
In this study, the authors reviewed the recent studies on CXCL12 signaling. Starting with the introduction of CXCL12 and its receptors CXCR4 and CXCR7/ACKR3, Mezzapelle et al. then discussed the lateral signaling of CXCR4 to the T cell receptor. Finally, the study about CXCR4 and CD47 was also mentioned. Overall, I think this review is interesting to some readers and my specific comments are listed below.
- I think the title “CXCL12 is a context-dependent signal in the crosstalk between tissues and immune cells” is irrelevant to the subject and should be improved. For example, the subtitle “3. Interactions of CXCR4 with ACKR3” is also confusing. It is hard to connect this content to the title.
- In figure 1, the icon of CXCL12 (orange ball) was shown. However, what’s the difference between the light orange ball and dark one?
- As far as I know, the CXCR12/ CXCR4 signaling is also existed in cancer cells, which is not shown in the figure 1. And CD47 should be shown in the icons just as CXCR4 and AKCR3.
- The abbreviations used in this manuscript should be expanded at first use. For convenience, please also provide a glossary in an appendix.
- It is better to show scalebar in figure 2 and quantify them.
